# SCALABLE EVALUATION OF CLOSED-SET AND OPEN-SET SEMANTIC AND SPATIAL ALIGNMENT IN LAYOUT-GUIDED DIFFUSION MODELS

## ABSTRACT

Evaluating layout-guided text-to-image generative models requires measuring both semantic alignment with textual prompts and spatial fidelity to prescribed layouts. Existing benchmarks are limited in scale and coverage, hindering systematic comparison and reducing interpretability of model capabilities. In this paper, we introduce a scalable closed-set benchmark (C-Bench), automatically built through a pipeline combining template- and LLM-based prompt generation with constraint-driven layout synthesis. C-Bench spans seven scenarios designed to isolate key generative capabilities and provides varying levels of complexity in both prompt structure and layout. To complement this controlled setting, we propose an open-set benchmark (O-Bench) derived from Flickr30k Entities, enabling evaluation on natural prompts and layouts. We further develop a unified evaluation protocol that combines semantic and spatial accuracy into a single score, enabling consistent model ranking. Using our benchmarks, we conduct a large-scale evaluation of six state-of-the-art layout-guided diffusion models, totaling 319,086 generated and evaluated images. Results show that MIGC achieves the highest overall performance (0.7082 on C-Bench and 0.7548 on O-Bench), establishing it as the most reliable model, particularly in layout alignment. Models trained explicitly with layout information consistently outperform Stable Diffusion–based approaches, which lag significantly behind. Overall, our benchmarks and evaluation protocol provide a scalable and interpretable framework for assessing progress in controllable image generation. Code and benchmarks are attached in Supplementary Materials.

## 1 INTRODUCTION

Recent advances in generative artificial intelligence have been driven by diffusion models, which now dominate text-to-image generation. These models are able to synthesize coherent and semantically faithful images from natural language prompts, and they are increasingly used in applications ranging from fashion image editing (Baldrati et al., 2023) to synthetic dataset generation (Parolari et al., 2024). In parallel, significant efforts have focused on fine-grained controllability (Li et al., 2023; Xie et al., 2023; Zhou et al., 2024), ensuring that the generated image also adheres to additional constraints such as layout, sketch or depth mask.

Among these, layout-guided generation has advanced rapidly due to its ease of use, yet evaluation has lagged behind (Grimal et al., 2024). Unlike standard text-to-image tasks, assessing these models requires considering two distinct but complementary dimensions: semantic alignment, which captures whether the correct objects appear in the image, and spatial alignment, which measures whether they are placed according to the prescribed layout. Without a standardized and scalable framework that accounts for both aspects, it is difficult to compare methods fairly or to track genuine progress.

An important first step in this direction was the introduction of 7Bench (Izzo et al., 2025) that, with respect to previous benchmarks (summarized in Tab. 1), introduced the evaluation of text *and* layout alignment. 7Bench is a benchmark made by prompt-layout pairs targeting a narrow set of challenges. It offers a systematic evaluation protocol for testing layout-guided diffusion models and makes an effort towards establishing a common ground for comparison. However, despite its value, the bench-

| Benchmark | Venue | #Scenario | #Instructions | Prompt | Layout |
|---|---|---|---|---|---|
| DrawBench (Saharia et al., 2022) | NeurIPS | 4 | 200 | T | ✗ |
| Visor (Gokhale et al., 2022) | arXiv | 1 | 25k | T | ✗ |
| HSR-Bench (Bakr et al., 2023) | ICCV | 13 | 45k | E | ✗ |
| TIFA v1.0 (Hu et al., 2023) | ICCV | 12 | 4k | T | ✗ |
| T2I-CompBench (Huang et al., 2023) | NeurIPS | 6 | 6k | T | ✗ |
| Layout-Bench (Cho et al., 2024) | CVPR | 1 | 8k | T+L | ✓ |
| 7Bench (Izzo et al., 2025) | ICPR | 7 | 224 | T | ✓ |
| **Ours** | **-** | **7+1** | **6.6k** | **T+L+E** | ✓ |

Table 1: Comparison of existing benchmarks. The column #Scenario indicates the number of categories while #Instructions shows the number of examples in benchmark. The Prompt column specifies whether prompts are template-based (T), generated through an LLM (L) or taken from existing datasets (E). The Layout column indicates whether instructions include bounding boxes.

mark suffers from significant limitations. Its small size, consisting of only 224 manually designed prompts, makes it overly sensitive to what the authors call "unlucky prompts" or ambiguous cases, leading to anomalous evaluation results. The manual design and annotation process makes scaling impractical, limiting extensions of the benchmark. Although human-annotated, its design constrains prompts to a narrow set of artificial patterns that lack the diversity and naturalness of human-written descriptions. Moreover, semantic and spatial fidelity are measured with separate metrics, which prevents the establishment of a clear and consistent ranking of models.

This work addresses these limitations with a three-fold contribution, also illustrated in Fig. 1. (i) We introduce a *scalable closed-set benchmark* (C-Bench) for systematically evaluating layout-guided text-to-image models under controlled conditions. The benchmark mimics 7Bench, but is automatically constructed through a pipeline that removes the need for costly manual annotations. This pipeline combines template- and LLM-based prompt generation with constraint-driven layout synthesis. Despite being automatically constructed, C-Bench covers varying levels of prompt and layout complexity at scale, analyzing specific generative capabilities such as object-attribute binding and spatial relations. (ii) To complement the controlled conditions of C-Bench, we introduce an *open-set benchmark* (O-Bench) derived from Flickr30k Entities (Plummer et al., 2017). O-Bench assesses models using natural prompts and real-world layouts, exposing them to the variability of human language and diverse object arrangements. This offers a realistic evaluation of model generalization in real-world settings. (iii) We introduce a unified evaluation protocol that combines semantic alignment and spatial fidelity into a single *unified score*. This metric enables comprehensive, interpretable assessment of generative models, supporting consistent ranking, direct comparison, and analysis of strengths and weaknesses in layout-guided image generation.

Using our benchmarks, we perform a large-scale evaluation of six state-of-the-art layout-guided diffusion models, totaling 319,086 generated and evaluated images. Our experiments demonstrate that MIGC delivers the strongest overall performance, making it the most reliable model, especially regarding layout alignment. Models trained explicitly with layout information consistently surpass Stable Diffusion-based approaches, indicating the importance of proper pre-training. Our analysis further shows that layout-guided generation performance decreases as the number of objects in the prompt increases, and that mixing multiple aspects raises task complexity and impacts accuracy. This aligns with previous work focusing solely on text alignment (Grimal et al., 2024). These findings emphasize the value of our comprehensive benchmarks and unified evaluation protocol, providing an interpretable framework for comparing and selecting models for downstream tasks. We hope these insights facilitate clearer evaluation of model outputs and support their use in generating synthetic datasets for domains where data are scarce, costly, or difficult to collect.

## 2 SCALABLE CLOSED-SET BENCHMARK

We construct the closed-set benchmark using an automatic pipeline that generates the instruction set employed to probe generative models. The pipeline consists of two main components. The Prompt Generation Engine (PGE) produces textual prompts by combining template-based rules and large language models. The Layout Generation Engine (LGE) then creates layouts through a constraint-

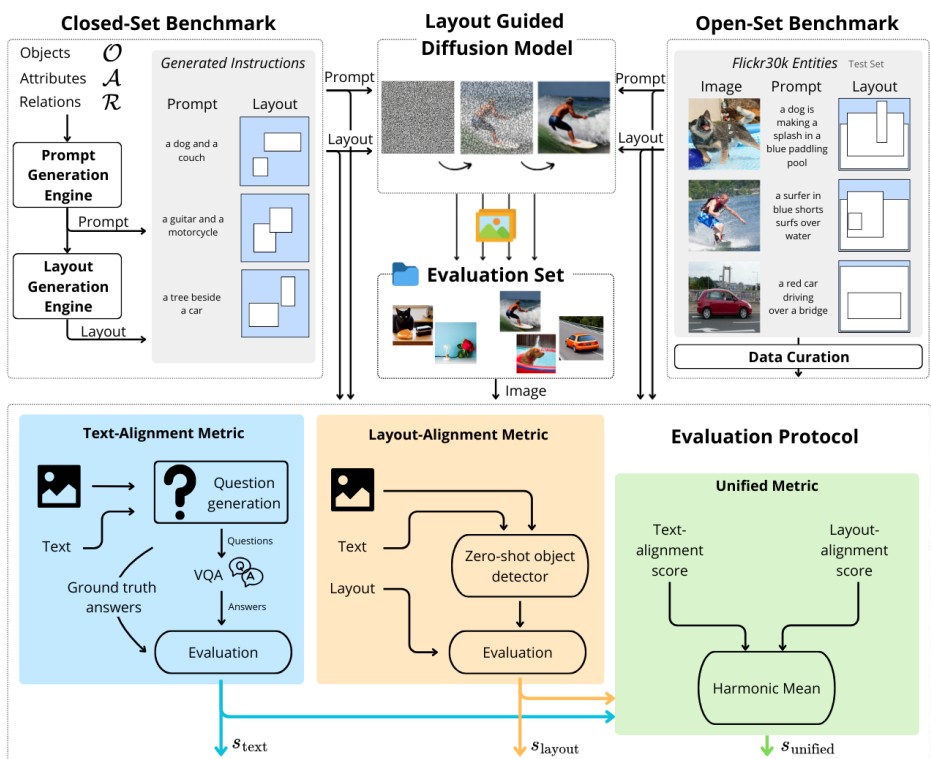

Figure 1: Overview of our evaluation framework. We introduce the closed-set benchmark, automatically built by generating prompts and layouts, and the open-set benchmark derived from Flickr30k Entities. A layout-guided model generates images from these prompts and layouts, forming the evaluation set. The evaluation protocol assesses the images providing text-alignment, layout-alignment and unified scores for both interpretability and consistent model ranking.

based procedure, arranging bounding boxes in a realistic manner while controlling randomness according to the content of each prompt. Despite its scalability, the pipeline preserves generality: the instruction set spans seven distinct scenarios (described later) and includes prompts with varying numbers of objects, thereby introducing structural and spatial complexity. An overview of the pipeline is depicted in Fig. 1 (top-left).

## 2.1 PROMPT GENERATION ENGINE

Textual prompts are generated using a combination of template-based rules and Large Language Models (LLMs). To ensure full control over the input and enable fair evaluation of outputs, we design six scenarios inspired by 7Bench (Izzo et al., 2025) with the goal to evaluate a specific generative capability. Specifically, we investigate:

*object biding*, i.e. the ability to generate elements described in the prompt;

*color binding*, i.e. the adherence of generated objects to color attributes;

*attribute binding*, i.e. the adherence to generic attributes like color, shape, material, appearance, and dimension;

*object relationship*, i.e. the ability to depict objects in relations, e.g. above, below, far from, to the left of, etc.;

*small bboxes*, i.e. the ability to depict objects whose size is between 3 and 10% of the image area;

*overlapped bboxes*, i.e. the ability to depict objects that overlaps in terms of layout.

For each scenario, we design a template that strictly constrain text structure, object-attribute combinations, and inter-object relations, thereby isolating the intended capability under evaluation. The

| Scenario | Template |
|---|---|
| Object binding | "$det(o_1)\ o_1,\ det(o_2)\ o_2,\ det(o_3)\ o_3$ and $det(o_4)\ o_4$" |
| Color binding | "$det(c_1)\ c_1\ o_1,\ det(c_2)\ c_2\ o_2,\ det(c_3)\ c_3\ o_3$ and $det(c_4)\ c_4\ o_4$" |
| Attribute binding | "$det(a_1)\ a_1\ o_1,\ det(a_2)\ a_2\ o_2,\ det(a_3)\ a_3\ o_3$ and $det(a_4)\ a_4\ o_4$" |
| Object relationship | "$det(o_1)\ \imath_{12}\ det(o_2)\ o_2$ and $det(o_3)\ \imath_{34}\ det(o_4)\ o_4$" |
| Small bboxes | "$det(o_1)\ o_1,\ det(o_2)\ o_2,\ det(o_3)\ o_3$ and $det(o_4)\ o_4$" |
| Overlapped bboxes | "$det(o_1)\ o_1,\ det(o_2)\ o_2,\ det(o_3)\ o_3$ and $det(o_4)\ o_4$" |

Table 2: Templates used by the Prompt Generation Engine for each scenario. Templates are shown with their full complexity, i.e. $N = 4$ objects.

templates are described with a formalism inspired by the disentangle representation theory (Trager et al., 2023), and are reported in Tab. 2. Each prompt describes up to four objects. For example, a template with $N= 2$ objects can be:

$$t = "det(o_1,a_1)\ a_1\ o_1\ rel(and, \imath_{12})\ det(o_2,a_2)\ a_2\ o_2" \tag{1}$$

where, $o_i \in \mathcal{O}$ the set of objects, $a_i \in \mathcal{A}$ the set of attributes, $det(o_i,a_i)$ is a determinant that depends on the object or attribute if present, and $rel(and, \imath_{ij})$ is the coordinating conjunction *and* or a relation $\imath_{ij} \in \mathcal{R}$–the set of relations–if present. To generate actual prompts, templates are instantiated by randomly picking objects, attributes and relations from their respective sets. We detail each set in Appendix A.3.

Beyond the template-driven cases, we introduce a seventh scenario: complex compositions, where all previously isolated challenges are combined into a single setting. Here, prompts are generated by an LLM conditioned on the same sets of objects, attributes, and relations. Unlike rigid templates, LLMs are well suited for this task, as they can produce natural, fluent, and contextually rich sentences while still adhering to the required constraints. Specifically, we employ few-shot learning (Brown et al., 2020) and provide instructions accompanied by example outputs. The LLM is given $\mathcal{O}$, $\mathcal{A}$, and $\mathcal{R}$ and instructed to combine them freely into coherent descriptions, while also guided to mention a specified number of objects (1-4), to enable fine-grained analysis. The full prompt template used for this scenario is reported in Appendix A.6.

## 2.2 LAYOUT GENERATION ENGINE

To collect the large number of bounding boxes associated to the prompts without relying on costly manual annotation, we designed a constraint-based Layout Generation Engine (LGE). It produces a reasonable layout for any given prompt. Specifically, we first obtain the set $\{o^i_j\}^N_{j=1}$ of objects described in the $i$-th prompt. In case this set is not provided by the Prompt Generation Engine (e.g. when prompts are LLM-generated), we extract the relevant objects automatically with a natural language parser such as spaCy (Honnibal et al., 2020). Then, for each object $o^i_j$, the LGE produces a bounding box: $b^i_j = [x^{\min}_j, y^{\min}_j, x^{\max}_j, y^{\max}_j]$ with $0 \leq x^{\min}_j < x^{\max}_j \leq 1$ and $0 \leq y^{\min}_j < y^{\max}_j \leq 1$.

Bounding box coordinates are chosen according to different constraints. Box size depends on the number of objects in the prompt: single-object prompts allow larger boxes (150-500 px per side), while two, three, and four objects use progressively smaller ranges (120-250, 100-180, and 80-150 px, respectively) to reduce clashes. For single-object prompts, the box is placed randomly within the image boundaries. For multi-object prompts, placement strategies vary: if spatial relations are involved (e.g., above, far from, to the left of), boxes are positioned accordingly. The first box is sampled randomly, leaving sufficient space for subsequent boxes to satisfy the relation; invalid placements trigger retries until a valid configuration is obtained. In case of overlapping scenarios, subsequent boxes are forced to overlap with at least one previously placed box. For prompts with multiple objects but no explicit relations, bounding boxes are placed randomly while enforcing non-overlap through rejection sampling, while complex composition does not follow any constraint.

## 3 OPEN-SET BENCHMARK

While the closed-set benchmark evaluates generative models under controlled conditions—using structured prompts and carefully designed bounding boxes—the open-set dataset is intended to as-

sess them in more natural settings. However, instead of relying on costly manual annotation of prompts and bounding boxes, we re-frame the purpose of Flickr30k Entities (Plummer et al., 2017), a widely used dataset in fine-grained visual-language tasks such as Visual Grounding (Rigoni et al., 2023). Flickr30k Entities provides real-world images paired with multiple human-annotated captions, each associated with bounding boxes localizing the mentioned objects. We carefully select these realistic prompts and unconstrained layouts to construct our open-set benchmark: this process is described in the next section and visually depicted in Fig. 1 (top-right).

## 3.1 DATA CURATION

We construct our open-set benchmark (O-Bench) on the Flickr30k Entities test set (Plummer et al., 2017), to avoid potential data leakage with the training sets of layout-guided text-to-image models. It comprises 4,969 captions which we down-sample to approximately two-thirds of its original size, resulting in 3,319 prompts. We down-sample the test set to avoid excessively large datasets, which would make evaluation impractical given the slow inference speed of current models. As later described in the evaluation setup, we generate 8 images per example with different random seeds to ensure robustness. This greatly increases computation time: for instance, BoxDiff (Xie et al., 2023) requires over 13 GPU-days to process the full test set, while the down-sampled version reduces this to 7.81 GPU-days.

Sampling was performed to preserve the original distribution of objects per sentence while discarding outlier prompts. Outliers, defined as prompts in the long tail of the distribution, were removed to reduce class imbalance, as they did not yield meaningful results. The final benchmark includes prompts describing different real-life situations including 1 to 8 objects, offering a realistic range for evaluation.

## 4 EVALUATION PROTOCOL

To evaluate the alignment between generated images and their input prompts, we adapt and extend the text-alignment score $s_{\text{text}}$ and the layout-alignment score $s_{\text{layout}}$ originally introduced in 7Bench (Izzo et al., 2025). Furthermore, we introduce a novel metric that provides a more comprehensive assessment by combining both dimension in one single score, thus, enabling precise ranking of models: the unified score $s_{\text{unified}}$. An overview is given in Fig. 1 (bottom).

## 4.1 TEXT-ALIGNMENT SCORE

The text-alignment score is defined as the TIFA score (Hu et al., 2023), a widely adopted measure of semantic consistency in text-to-image generation. TIFA quantifies alignment as the proportion of correct responses provided by a Vision Question Answering (VQA) model when analyzing the generated images. To perform the evaluation, a set of questions and corresponding answers are automatically derived from the input prompt using a Large Language Model (LLM), ensuring independence from the image generation process. The VQA model is then queried on the generated image, and its responses are compared against the expected answers to compute the final score. Formally, let $\mathcal{X} = \{(T_i, I_i)\}_{i=1}^{N}$ be the set of input text prompts and generated images. For each example $i$, a set of questions, expected answers and actual answer is obtained: $\{Q_{j,i}, A_{j,i}, A_{j,i}^{\text{VQA}}\}_{j=i}^{M_i}$. The layout alignment score for example $i$ is defined as the average accuracy of VQA answers over expected answers. The overall score is then obtained as the mean across all examples:

$$s_{\text{text}}(i) = \frac{1}{M_i} \sum_{j=1}^{M_i} \mathbf{1}[A_{j,i}^{\text{VQA}} = A_{j,i}] \qquad s_{\text{text}} = \frac{1}{N} \sum_{i=1}^{N} s_{\text{text}}(i). \tag{2}$$

To better understand model behavior, we extend the base score $s_{\text{text}}$ by conditioning the evaluation on two axes: the scenario and the number of objects in the prompt. Analyzing the score along one or the other enhances interpretability in the evaluation process. Let $\mathcal{C}$ be the set of scenarios (e.g., object binding, color binding, etc), and $\mathcal{D} = \{1, 2, 3, 4\}$ the number of objects in the prompt. Let $\sigma(i) \in \mathcal{C}$ be the scenario of example $i$, and $\nu(i) \in \mathcal{C}$ its number of objects. We obtain the set of

indices $\mathscr{Z}_{c,d} = \{i \in \{1, \ldots, N\} : \sigma(i) = c, \ \nu(i) = d\}$ and overload the definition of $s_{\text{text}}$:

$$s_{\text{text}}^{c,d} = \frac{1}{|\mathscr{Z}_{c,d}|} \sum_{i \in \mathscr{Z}_{c,d}} s_{\text{text}}(i), \qquad s_{\text{text}}^{d} = \frac{1}{|\mathscr{Z}_{d}|} \sum_{i \in \mathscr{Z}_{c,d} \forall c} s_{\text{text}}^{c,d}, \qquad s_{\text{text}}^{c} = \frac{1}{|\mathscr{Z}_{c}|} \sum_{i \in \mathscr{Z}_{c,d} \forall d} s_{\text{text}}^{c,d}. \quad (3)$$

For example, $s_{\text{text}}^{2}$ is the text-alignment score on all examples with exactly two objects in the prompt from any scenario, while $s_{\text{text}}^{\text{object binding}}$ is the text-alignment score on all examples from object binding scenario and any number of objects in the prompt.

## 4.2 LAYOUT-ALIGNMENT SCORE

We define the layout-alignment score $s_{layout} \in [0, 1]$ as the Area Under Curve (AUC) of accuracy@k values computed over a range of Intersection over Union (IoU) thresholds. The $s_{layout}$ score captures the spatial accuracy of objects' placement within the generated images.

Specifically, let $\mathcal{X} = \{(T_i, O_i, I_i)\}_{i=1}^{N}$, be the set of input text prompts, objects with locations and generated images. Each prompt describe $M_i$ objects ($1 \leq M_i \leq 4$), and for each object $o_j^i$ we are also gives its target position as a bounding box $t_j^i$. Through an object detector, such as OWLv2 (Minderer et al., 2023), we obtain a set of $K$ detections associated with their corresponding label and confidence score for each example $i$: $D^i = \{(d_k^i, l_k^i, c_k^i)\}_{k=1}^{K}$. For each $j$, we filter the set of detections by matching the label with the object, thus $D_j^i = \{(d_k^i, l_k^i, c_k^i) \in D_i \mid l_k^i = o_j^i\}$. Among the filtered detections $D_j^i$, we select the bounding box with the higher confidence score $\hat{d}_j^i = \arg\max_{(d_k^i, l_k^i, c_k^i) \in \hat{D}_j^i} c_k^i$. We then compute the IoU between the selected detection $\hat{d}_j^i$ and the target bounding box $t_j^i$, denoted as $\text{IoU}_j^i = \text{IoU}(t_j^i, \hat{d}_j^i)$. Subsequently, these IoU values are thresholded at multiple levels $k \in \{0, 0.1, \ldots, 1\}$ to calculate $\text{Acc@}k_j^i = \frac{1}{M_i} \sum_{j=1}^{M_i} \mathbf{1}[\text{IoU}_j^i \geq k]$. The layout-alignment score for the single example is the area under the resulting $\text{Acc@}k$ curve, and the overall score is averaged across all examples:

$$s_{\text{layout}}(i) = \frac{1}{M_i} \sum_{j=1}^{M_i} \text{AuC}(\text{Acc@}k_j^i), \qquad s_{\text{layout}} = \frac{1}{N} \sum_{i=1}^{N} s_{\text{layout}}(i). \quad (4)$$

Similarly to the text-alignment score, we extend the formulation to evaluate along different axis: scenario $s_{\text{layout}}^{c}(i)$, number of object $s_{\text{layout}}^{d}(i)$ and both $s_{\text{layout}}^{c,d}(i)$.

## 4.3 UNIFIED SCORE

Ranking the performance of layout-guided text-to-image models poses a complex challenge, as it requires capturing both semantic alignment with the input text and spatial fidelity to the given layout. While text and layout alignment enable fine-grained interpretability of the generation capabilities, a unified metric that accounts for both aspects is necessary to compare and rank different models. For this reason, we propose the unified score, $s_{\text{unified}}$, as the combination of $s_{\text{text}}$ and $s_{\text{layout}}$:

$$s_{\text{unified}} = H(s_{\text{text}}, s_{\text{layout}}) = \frac{2 \cdot s_{\text{text}} \cdot s_{\text{layout}}}{s_{\text{text}} + s_{\text{layout}}}. \quad (5)$$

The harmonic mean penalizes imbalances between the two components, ensuring that strong performance requires both textual and spatial consistency. This choice is inspired by its widespread use in multi-objective evaluation settings, such as the F1 score in information retrieval (Blair, 1979).

Similarly to text and layout alignment scores, we generalize $s_{\text{unified}}$ for the analysis along scenarios, number of objects or both axis: $s_{\text{unified}}^{c}(i)$, $s_{\text{unified}}^{d}(i)$ and $s_{\text{unified}}^{c,d}(i)$.

## 5 EXPERIMENTAL EVALUATION

### 5.1 SETUP

**C-Bench** Following the pipeline described in Sec. 2, we generated the closed-set benchmark (C-Bench), resulting in 3,328 instructions. C-Bench is more than $14\times$ larger than 7Bench, which

| Model | Training-free | Base model | Throughput (secs per image) | Memory VRAM (GB) |
|---|---|---|---|---|
| SD v1.4 (Rombach et al., 2022) | ✗ | - | 7.62 | 6.25 |
| CA Guidance (Chen et al., 2024) | ✓ | SD v1.4 | 19.98 | 6.27 |
| GLIGEN (Li et al., 2023) | ✗ | - | 12.50 | 7.13 |
| Attn. Refocus (Phung et al., 2024) | ✓ | GLIGEN | 28.26 | 22.77 |
| BoxDiff (Xie et al., 2023) | ✓ | GLIGEN | 20.66 | 28.26 |
| MIGC (Zhou et al., 2024) | ✗ | - | 9.40 | 8.84 |

Table 3: List of models under test. Throughput and memory footprint have been measured on a node with an Intel Xeon processor, 60GB of RAM and one Nvidia RTX A6000 GPU.

contains only 224 samples evenly distributed across the seven scenarios. Thanks to the scalability of our pipeline, C-Bench ensures a balanced number of prompts per scenario, maintains a fair distribution of object occurrences, and provides comprehensive coverage of bounding box sizes without requiring costly manual annotations.

From a technical standpoint, each sample in C-Bench consists of a textual prompt paired with a set of bounding boxes, one for each object described. Furthermore, for every object, the benchmark also provides a noun phrase that specifies the entity together with its attributes and qualifiers. Prompts for the complex composition scenario are generated by ChatGPT (Hurst et al., 2024). The benchmark organized as a CSV file is attached in the Supplementary Materials.

**Models Under Test**    We evaluate 6 popular layout-guided text-to-image diffusion models on our closed and open-set benchmarks. Models, summarized in Tab. 3, are all open source and have been accurately chosen to explore a wide range of methods and techniques. In particular, we test GLIGEN (G) (Li et al., 2023) and MIGC (M) (Zhou et al., 2024), which are trained with layout information, and three training-free approaches: Attention Refocusing (G_AR) (Phung et al., 2024), BoxDiff (G_BD) (Xie et al., 2023), and Cross Attention Guidance (SD_CAG) (Chen et al., 2024). The first two are built on top of GLIGEN, the last one uses Stable Diffusion as the underlying model. Finally, we include Stable Diffusion v1.4 (SD) (Rombach et al., 2022) in the analysis for a comparison in terms of textual alignment.

**Evaluation Setting**    Following previous works (Grimal et al., 2024; Izzo et al., 2025), we generate 8 images for each instruction in the benchmark, varying the seed from 1 to 8 to ensure robustness against sampling variability and to obtain a reliable estimate of model performance. Across all models under test, the generation produced a total of 159,744 and 159,312 images for C-Bench and O-Bench, respectively. Each image was generated with a resolution of $512 \times 512$ We evaluate the generated images using the evaluation protocol described in Section 4. We use pre-trained weights for TIFA [1] and OWLv2[2].

## 5.2 RESULTS

We present the ranking on both closed and open set benchmarks according to our novel $s_{\text{unified}}$ in Tab. 4. The unified score provides a single, comprehensive measure of performance, allowing models to be ranked consistently by their overall semantic and spatial alignment. From our evaluation MIGC achieves a unified score of 0.7082 and 0.7548 on C-Bench and O-Bench respectively, signaling its robustness in both semantic and spatial aspects of generation. GLIGEN-based models also obtain decent performance (0.6070-0.6537, 0.7305-0.7517), showing the importance of pre-training with layout information. Instead, Cross-Attention Guidance–which is based on Stable Diffusion–obtains a unified score of 0.3747 and 0.5370: a considerable drop in performance with respect to top-performing model. The following sections present a detailed breakdown of the results, offering deeper interpretability of the unified score and clearer insights into the specific capabilities and limitations of each model.

---

[1]https://github.com/Yushi-Hu/tifa
[2]https://huggingface.co/docs/transformers/model_doc/owlv2

| Model | Closed-set | | Open-set | |
|---|---|---|---|---|
| | $s_{\text{unified}}$ | $\Delta\%$ | $s_{\text{unified}}$ | $\Delta\%$ |
| MIGC (Zhou et al., 2024) | **0.7082** | +0.0 | **0.7548** | +0.0 |
| BoxDiff (Xie et al., 2023) | 0.6537 | −8.4 | 0.7410 | −1.8 |
| GLIGEN (Li et al., 2023) | 0.6143 | −13.3 | 0.7517 | −0.1 |
| Attention Refocusing (Phung et al., 2024) | 0.6070 | −14.3 | 0.7305 | −3.2 |
| Cross-Attention Guidance (Chen et al., 2024) | 0.3747 | −47.1 | 0.5370 | −28.9 |
| Stable Diffusion 1.4* (Rombach et al., 2022) | 0.2522 | −64.4 | 0.4505 | −40.3 |

Table 4: Ranking for the models evaluated on the closed-set benchmark. *Stable Diffusion does not have layout capabilities. **Bold** = best model, Underline = second best. $s_{\text{unified}}$ represent our novel unified score and $\Delta\%$ is the performance delta between model's score and top performer.

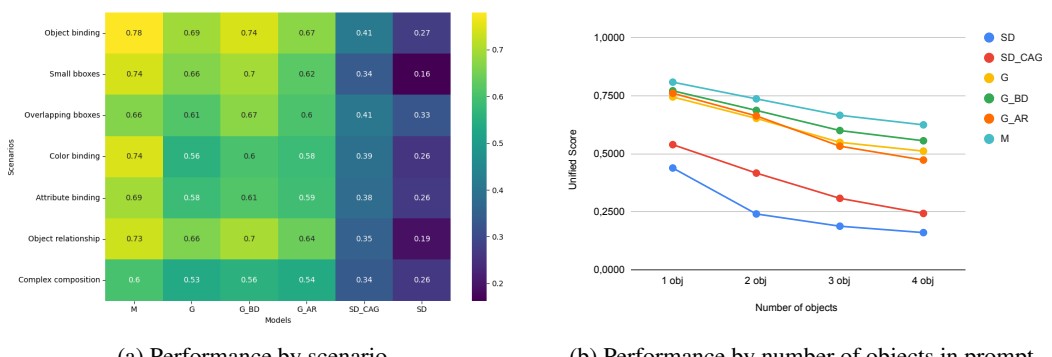

(a) Performance by scenario.          (b) Performance by number of objects in prompt.

Figure 2: Performance breakdown by scenario and object count, measured with the unified score for the six models under test.

## 5.3 CLOSED-SET RESULTS BREAKDOWN

**Breakdown by Scenario and Object Count**   To further interpret the ranking and provide a more detailed view of model performance, we also report results broken down by scenario and by the number of objects in the prompt. Fig. 2a confirms the overall ranking, with MIGC achieving the highest scores across almost all scenarios. As expected, the complex composition scenario proves more challenging than the others, with smaller performance gaps among models. Fig. 2b additionally illustrates how performance decreases as the number of objects in the prompt increases. All models exhibit a decline in accuracy when handling more objects. Interestingly, aside from Stable Diffusion–which does not support layout input–models show a similar degradation pattern, suggesting comparable sensitivity to object count.

**Breakdown on Text and Layout Alignment**   Fig. 3 and Fig. 4 report both text-alignment and layout-alignment scores, revealing that the primary source of errors stems from layout alignment. MIGC demonstrates strong performance in layout adherence across all scenarios. Interestingly, while Cross-Attention Guidance (SD_CAG) performs well on $s_{\text{text}}$, its performance drops sharply for $s_{\text{layout}}$. This drop explains why, as shown in the previous section, our unified metric $s_{\text{unified}}$ ranks SD_CAG fifth out of six models, effectively penalizing good text alignment when accompanied by poor layout fidelity. This trend is consistent also when analyzing the results by number of objects, confirming that layout-aware models maintain an advantage even at higher complexity. While text and layout alignment alone are not sufficient to evaluate overall performance, they provide valuable interpretability into the sources of errors.

## 5.4 OPEN-SET RESULTS BREAKDOWN

To complement the global ranking, we also report results broken down by object count on the open-set benchmark. We recall that the open-set benchmark includes prompts describing between 1 and 8

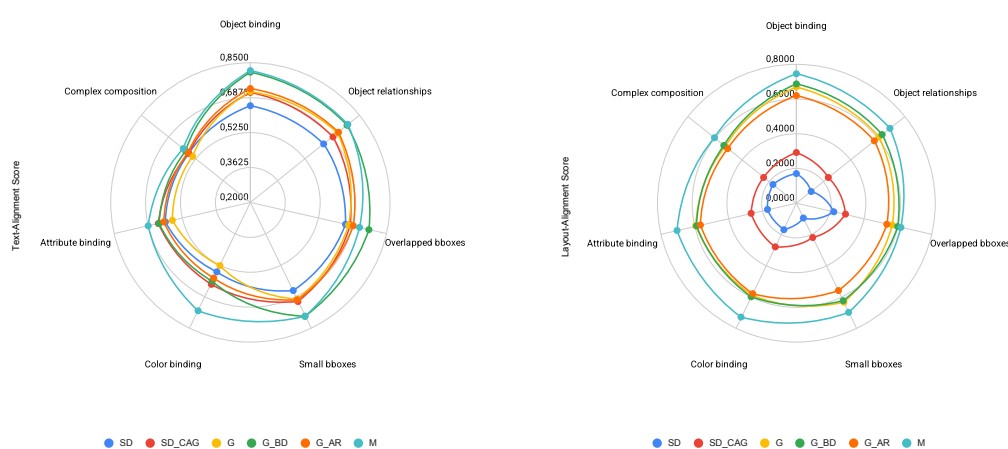

Figure 3: Performance breakdown on scenarios, measured by text-alignment score (left) and layout-alignment score (right).

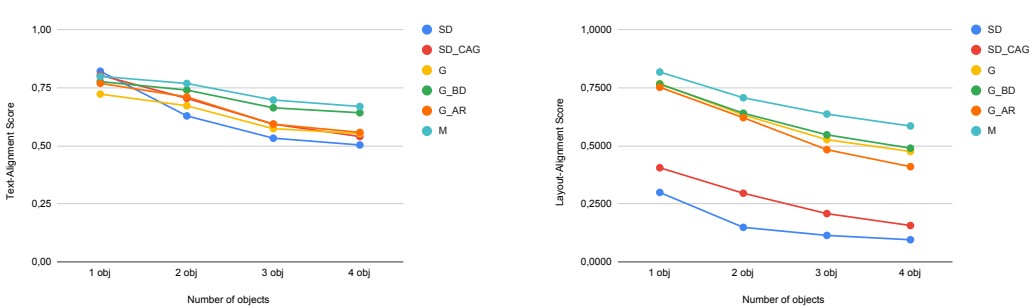

Figure 4: Performance breakdown on number of objects in the prompt, measured by text-alignment score (left) and layout-alignment score (right).

objects. Fig. 5 confirms expectations: model performance decreases as the number of objects in the prompt increases. Similar to the closed-set benchmark, all models exhibit a comparable degradation pattern, indicating consistent sensitivity to object count.

# 6 CONCLUSION

We presented a comprehensive framework for evaluating layout-guided text-to-image generative models. By introducing a scalable closed-set benchmark, an open-set benchmark grounded in natural data, and a unified evaluation protocol, we enable systematic, fair, and reproducible assessment of model performance. We hope this work provides researchers with the tools to interpret, rank, and evaluate these models, enabling clearer assessment of their outputs, informing their use in applications like synthetic dataset generation, and tracking genuine progress in the field.

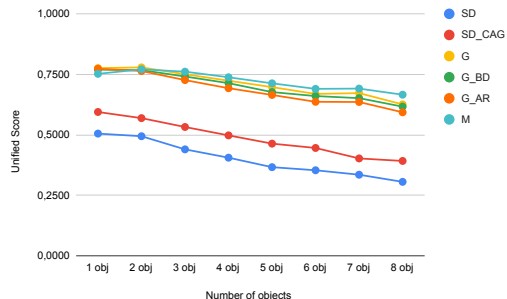

Figure 5: Performance breakdown by object count on open-set benchmark, measured with the unified score.

REPRODUCIBILITY STATEMENT

To ensure reproducibility, we provide detailed descriptions of benchmarks, pipeline, and evaluation protocols in both the main paper and Appendix. Specifically, we describe the pipeline to obtain C-Bench in Sec. 2 and detailed the set of object $\mathcal{O}$, attributes $\mathcal{A}$ and relations $\mathcal{R}$ used for templates in Appendix A.3. We also report the full custom prompt used to obtain the complex composition instructions in Appendix A.6. The O-Bench benchmark is detailed in Sec. 3. We include extra analysis on the two benchmarks in Appendix A.1 and A.2, and visualize qualitative examples of instructions and generated images in Appendix A.4 and A.5. All the details for the evaluation protocol are described in Sec. 4. Code for obtaining both benchmarks, running and evaluating the layout-guided generative models, as well as the benchmarks themselves are attached in the Supplementary Material and will be made publicly available upon acceptance. In the code we provide documentation with installation and usage instructions for the models as well as the evaluation protocol.

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

## A APPENDIX

### A.1 OBJECTS AND BOUNDING BOXES DISTRIBUTION IN C-BENCH

To better understand the characteristics of C-Bench, we provide an analysis of its object and layout distributions. Fig. 6a reports the frequency of objects across all prompts, highlighting the coverage and balance of categories represented in the benchmark. Fig. 6b shows the distribution of bounding box areas for each scenario, illustrating how object sizes vary across different settings. Together, these analyses confirm that C-Bench offers both semantic diversity and spatial variability, while maintaining controlled conditions for systematic evaluation.

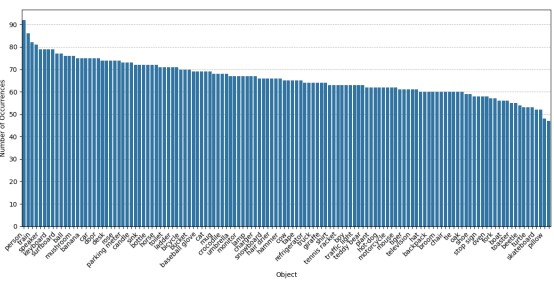
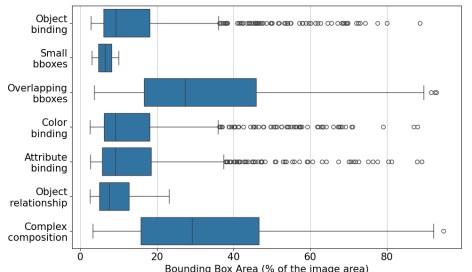

(a) Frequency of objects across all prompts on C-Bench.

(b) Distribution of bounding boxes area per scenario. Values are displayed percentage with respect to the image size.

Figure 6: Analysis of the C-Bench. On the left we report the frequency of objects across all prompts, while on the right we show the distribution of bounding box areas per scenario.

| rose | oak | beetle | skyscraper | tree |
| baby | bed | lamp | dog | laptop |
| bicycle | person | car | bus | cat |
| book | chair | boy | couch | table |
| plant | toilet | cellphone | microwave | sheep |
| boat | banana | stop sign | donut | cow |
| clock | bottle | umbrella | bird | guitar |
| toothbrush | parking meter | bench | platypus | keyboard |
| baseball bat | vase | surfboard | tiger | train |
| flower | sandwich | spoon | pizza | carrot |
| teddy bear | hot-dog | skateboard | kite | broom |
| apple | handbag | horse | snowboard | giraffe |
| tie | shower | traffic light | bear | toaster |
| knife | baseball glove | crocodile | suitcase | fork |
| cake | cup | bowl | hair drier | elephant |
| mouse | mushroom | motorcycle | turtle | tennis racket |
| truck | zebra | fire hydrant | oven | sink |
| frisbee | hat | ruler | shoe | ball |
| candle | ladder | charger | mug | tape |
| shirt | pillow | pan | plate | shampoo |
| hammer | blender | basket | screwdriver | wallet |
| bin | leaf | bucket | monitor | watch |
| flashlight | sock | door | scarf | speaker |
| desk | backpack | printer | remote | glass |
| curtain | toolbox | drill | notebook | television |
| soap | ring | refrigerator | | |

Table 5: List of objects $\mathcal{O}$ used by the Prompt Generation Engine to instantiate a template.

## A.2  OBJECT DISTRIBUTION IN O-BENCH

Fig. 7 reports the distribution of prompts in O-Bench with respect to the number of objects they describe. The dataset naturally exhibits variability in object counts, ranging from 1 to 8 objects in single prompt. This analysis highlights the diversity of prompts and motivates the use of O-Bench as a realistic benchmark for evaluating model performance under different levels of compositional complexity.

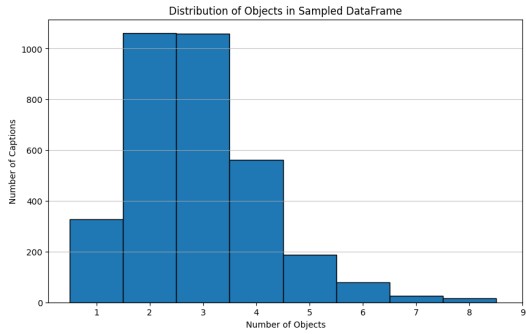

Figure 7: Number of prompts per object count on O-Bench.

## A.3  OBJECT, ATTRIBUTE, AND RELATION SETS

For template-based prompt generation in C-Bench, we define three sets: objects $\mathcal{O}$, attributes $\mathcal{A}$, and relations $\mathcal{R}$. We detail each of them in Tab. 5, Tab. 6 and Tab. 7 respectively.

| aggressive | dark | large | short | tall |
| black | fast | pink | silver | warm |
| blue | fluffy | red | small | white |
| bright | fuzzy | rotten | smooth | wooden |
| clean | green | rough | snowy | yellow |
| crowded | happy | shiny | soft | |

Table 6: List of attributes $\mathscr{A}$ used by the Prompt Generation Engine to instantiate a template.

| above | far from | next to | over | to the right of |
| below | near | on | under | to the left of |
| beside | | | | |

Table 7: List of relations $\mathscr{R}$ used by the Prompt Generation Engine to instantiate a template.

### A.4 QUALITATIVE EXAMPLES OF INSTRUCTIONS

We present qualitative examples of the instruction set in Fig 8, consisting of paired prompts and layouts. These examples illustrate the diversity and complexity of scenarios included in C-Bench, ranging from simple single-object cases to more challenging multi-object compositions involving attributes and spatial relations, as well as natural prompts in O-Bench. Each instruction defines both the textual description and the corresponding spatial arrangement, ensuring precise control over what is generated and where it should appear. The first figure showcases representative samples across different scenarios and object counts, highlighting how the benchmark systematically isolates key generative challenges while maintaining scalability and interpretability.

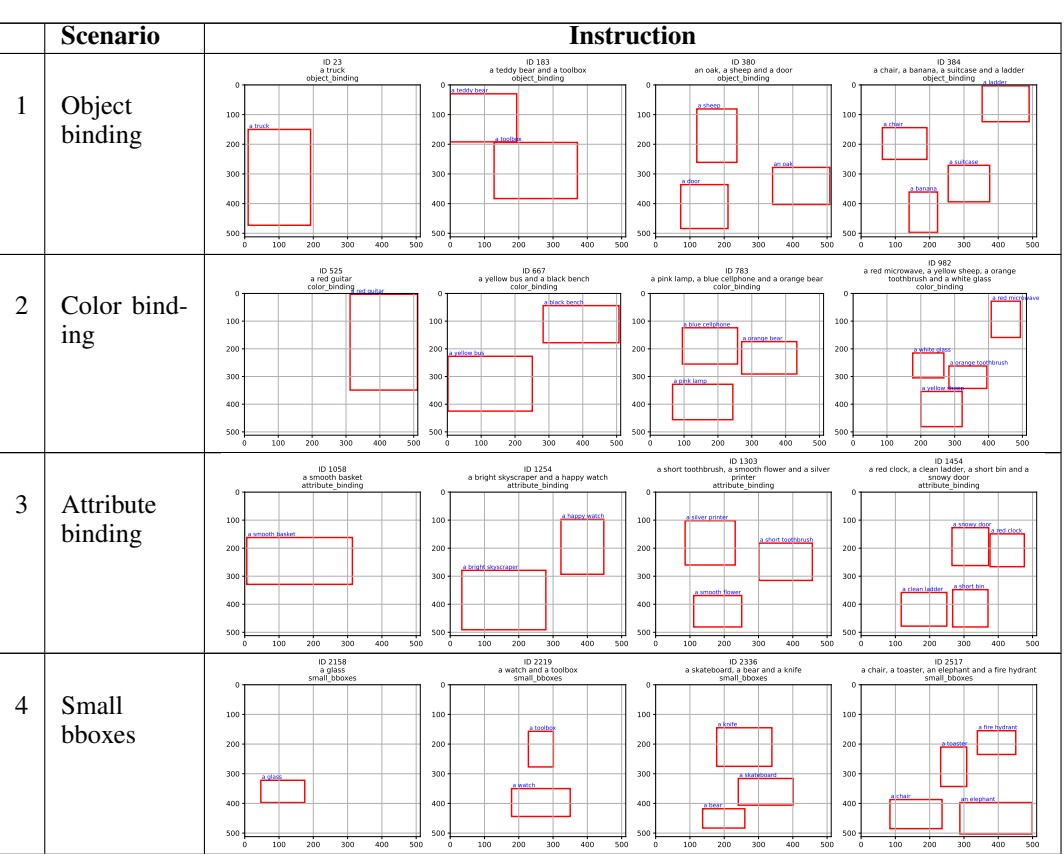

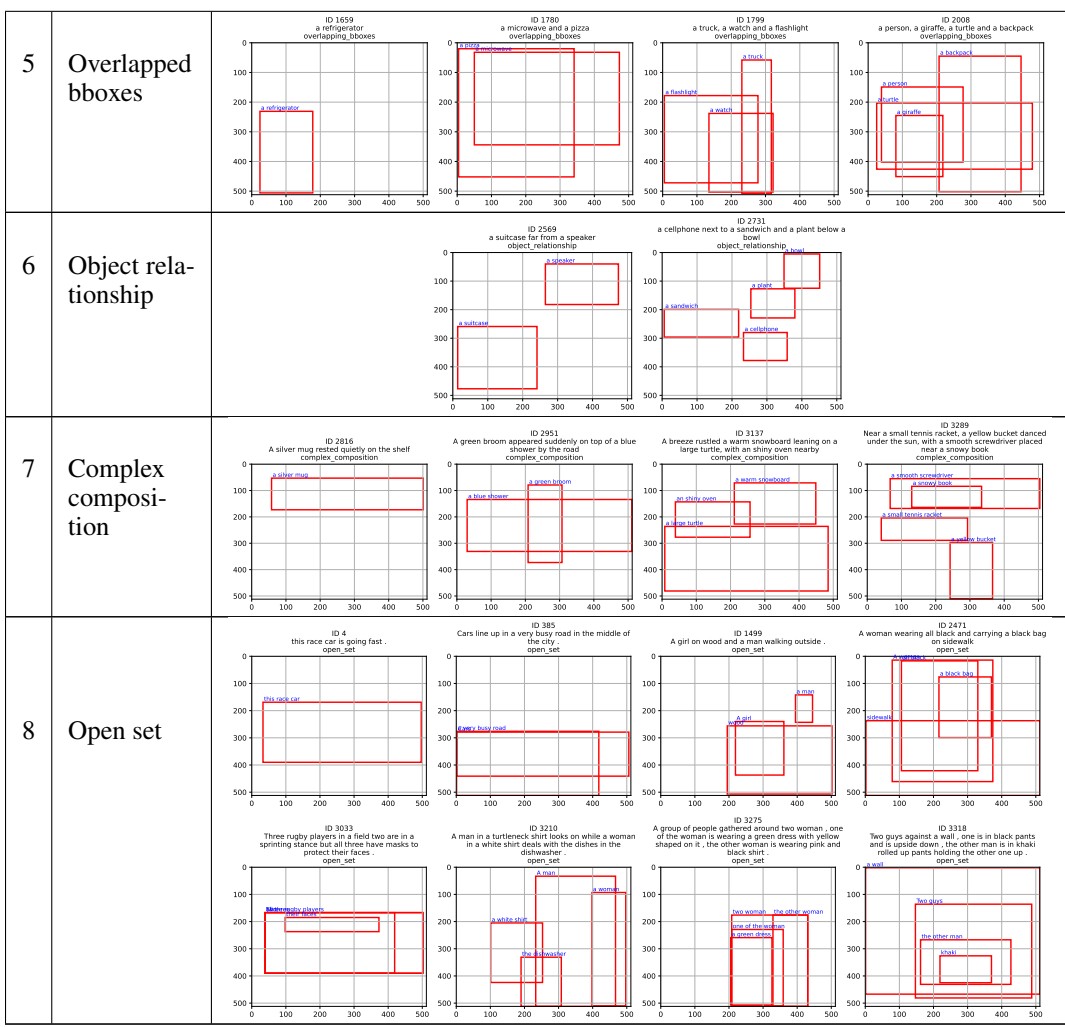

Table 8: Examples of instructions, i.e. prompt-layout pairs, from both C-Bench (rows 1-7) and O-Bench (row 8).

## A.5 QUALITATIVE EXAMPLES OF GENERATED IMAGES

To provide a qualitative perspective on model behavior, we include additional examples of generated images in Tab. 9. The figure shows, for each scenario in C-Bench successful generations, where objects are correctly rendered with the expected attributes and spatial configuration as well as failure cases, which highlight common challenges such as missing objects, incorrect attributes, or misplaced elements.

Similarly, we

| | Scenario | Prompt | Success | Fail |
|---|---|---|---|---|

| 1 | Object binding | A bus, a snowboard and a bowl. |  |  |
|---|---|---|---|---|
| 2 | Small bboxes | A bench, a truck and a candle. |  |  |
| 3 | Overlapping bboxes | A book, a scarf and a soap. |  |  |
| 4 | Color binding | A pink oak, a brown couch and a yellow speaker. |  |  |
| 5 | Attribute binding | A wooden train and a clean plate. |  |  |
| 6 | Object relationship | A refrigerator beside a tree. |  |  |

| 7 | Complex composition | A yellow bus stood above a rough crocodile in the quiet morning. |  |  |
|---|---|---|---|---|
| 8 | Open set | A soccer player is running while kicking a ball. |  |  |
| 9 | Open set | A baby boy in overalls is crying. |  |  |

Table 9: Examples of generated images from both C-Bench (rows 1-7) and O-Bench (rows 8-9). The Success column illustrates correct generations, where all objects appear with the intended attributes, while the Fail column shows cases where generation errors occur.

## A.6 CUSTOM LLM PROMPT FOR GENERATION OF COMPLEX COMPOSITIONS

As described in Sec. 2, we used large language models, specifically ChatGPT (Hurst et al., 2024), to generate prompts for the complex composition scenario, where all previously isolated challenges are combined into a single setting. The listing below shows the custom prompt used for $N = 4$ objects per sentence, which can be easily adapted to cases with 1–3 objects. We executed this prompt four times to obtain the complex compositions for each case.

```
Generate 128 natural compositional phrases with various structures and
    creativity.
Each prompt must describe a scene involving exactly 4 unique objects.
    Objects can be reused across multiple prompts, but each object must
    appear only once within any given prompt. Each object must be
    enriched with at least one descriptive attribute, which may describe:

Color, shape, material, appearance, or dimension

Or a spatial relation between objects in the same prompt

Each prompt should be a short, vivid sentence that situates the objects
    in a dynamic or descriptive context, similar to the following
    examples:

A bright pink flower swayed gently under the tall oak tree.

A black laptop rested beside a green coffee mug on the messy desk.
```

```
Avoid passive or overly generic constructions-aim for imaginative,
    specific scenarios.

Adjust the articles of the objects if needed.

Object List (You can freely reuse any of these objects across different
    prompts, but not within the same prompt.)

obj_with_articles = [
 'a rose', 'an oak', 'a beetle', 'a skyscraper', 'a tree', 'a baby', 'a
    bed', 'a lamp', 'a dog', 'a laptop', 'a bicycle', 'a person', 'a car
    ', 'a bus', 'a cat', 'a book', 'a chair', 'a boy', 'a couch', 'a
    table', 'a plant', 'a toilet', 'a cellphone', 'a microwave', 'a
    sheep', 'a boat', 'a banana', 'a stop sign', 'a donut', 'a cow', 'a
    clock', 'a bottle', 'an umbrella', 'a bird', 'a guitar', 'a
    toothbrush', 'a parking meter', 'a bench', 'a platypus', 'a keyboard
    ', 'a baseball bat', 'a vase', 'a surfboard', 'a tiger', 'a train',
    'a flower', 'a sandwich', 'a spoon', 'a pizza', 'a carrot', 'a teddy
     bear', 'an hot-dog', 'a skateboard', 'a kite', 'a broom', 'an apple
    ', 'a handbag', 'a horse', 'a snowboard', 'a giraffe', 'a tie', 'a
    shower', 'a traffic light', 'a bear', 'a toaster', 'a knife', 'a
    baseball glove', 'a crocodile', 'a suitcase', 'a fork', 'a cake', 'a
     cup', 'a bowl', 'a hair drier', 'an elephant', 'a mouse', 'a
    mushroom', 'a motorcycle', 'a turtle', 'a tennis racket', 'a truck',
     'a zebra', 'a fire hydrant', 'an oven', 'a sink', 'a frisbee', 'a
    hat', 'a ruler', 'a shoe', 'a ball', 'a candle', 'a ladder', 'a
    charger', 'a mug', 'a tape', 'a shirt', 'a pillow', 'a pan', 'a
    plate', 'a shampoo', 'a hammer', 'a blender', 'a basket', 'a
    screwdriver', 'a wallet', 'a bin', 'a leaf', 'a bucket', 'a monitor
    ', 'a watch', 'a flashlight', 'a sock', 'a door', 'a scarf', 'a
    speaker', 'a desk', 'a backpack', 'a printer', 'a remote', 'a glass
    ', 'a curtain', 'a toolbox', 'a drill', 'a notebook', 'a television
    ', 'a soap', 'a ring', 'a refrigerator'
]

Allowed Attributes (for appearance or material):

attributes = [ 'aggressive', 'black', 'blue', 'bright', 'clean', 'crowded
    ', 'dark', 'fast', 'fluffy', 'fuzzy', 'green', 'happy',
'large', 'pink', 'red', 'rotten', 'rough', 'shiny', 'short', 'silver', '
    small', 'smooth', 'snowy', 'soft', 'tall', 'warm', 'white', 'wooden',
    'yellow'
]

Allowed Colors (subset of attributes):

colors = ['black', 'blue', 'brown', 'gray', 'green', 'pink', 'purple', '
    red', 'white', 'yellow', 'orange']

Spatial Relations (to be used as part of attributes or composition logic)
    :

spatial\_relations = [
'on top of', 'beside', 'under', 'above', 'next to', 'beneath', 'behind',
    'in front of', 'between', 'leaning on',
'inside', 'resting on', 'attached to', 'surrounded by', 'placed near'
]

Output Format:

Return the result as a CSV with two columns:

prompt, object1, object2, ..

Each row should contain:
```

```
The generated sentence

The noun chunks used

Example output format (for N=2):

prompt,object1,object2

A fluffy cat jumped onto the soft couch near the window" ,a fluffy cat, a
    soft couch

A red bicycle leaned against a wooden bench in the park ,a red bicycle, a
    wooden bench

The prompts should be inside quotes if needed to keep the correct number
    of columns in the CSV.

In the sentence, keep noun chunks unbroken, adjectives modifying a noun
    should not be split by commas. Treat the entire noun chunk as a
    single unit (e.g., "a small red ball", not "a small, red ball").

The articles should remain consistent between the prompt and the noun
    chunks.
```

## A.7 Use of Large Language Models

We used ChatGPT-5 free as an assistive tool to improve the clarity, conciseness, and overall readability of the manuscript. Specifically, the model was employed to refine sentence structure, enhance paragraph flow, rephrase sections for scientific style, and suggest more concise formulations without altering the scientific content or results. All conceptual contributions, experimental design, data collection, analysis, and interpretations were performed solely by the authors. The LLM acted strictly as a writing and language support tool and did not contribute to the research ideas, experimental methodology, or scientific findings.