# OpenReview forum: "Scalable Evaluation of Closed-Set and Open-Set Semantic and Spatial Alignment in Layout-Guided Diffusion Models"
_ICLR.cc/2026/Conference — ICLR 2026 Conference Withdrawn Submission_

### Official Review · Reviewer_7a9j · 2025-10-26

**Soundness:** 2
**Presentation:** 2
**Contribution:** 1
**Rating:** 4
**Confidence:** 4

**Summary:**

This paper introduces C-Bench, a scalable automatically generated closed-set benchmark, and O-Bench, an open-set benchmark based on Flickr30k Entities, for evaluating layout-guided text-to-image diffusion models. It also proposes a unified evaluation score combining semantic and spatial alignment. The empirical evaluation across six diffusion models provides comprehensive results and reproducible protocols.

**Strengths:**

The automated pipeline for generating C-Bench (3,328 samples vs. 224 in 7Bench) addresses a real limitation in current benchmarks and reduces manual annotation costs.

The dual benchmark approach (closed-set for controlled evaluation, open-set for realistic scenarios) is well-motivated and provides complementary perspectives.

The harmonic mean combining semantic and spatial alignment provides a principled way to rank models, though this choice could be better justified.

319086 generated images across 6 models represents substantial empirical work.

**Weaknesses:**

## From the perspective of the work itself

Poor contribution. Although this work is well-presented, it lacks methodological novelty:
- There are 6 scenarios but they are exactly the same with 6 in 7 scenarios in 7Bench. And the 7th scenario for template-driven cases “complex compositions” is also proposed by 7Bench. The paper uses “design” and “introduce” those scenarios, which is not that proper.
- The extensions for text-alignment and layout-alignment scores are limited to the way of expression and convenience for recognition. The text-alignment score is directly adopted from TIFA, and The layout-alignment score is adapted from 7Bench with minor modifications.
- The harmony mean is justified only intuitively. Why not weighted mean, why not geometric mean? There is no empirical or human validation to verify whether this measure correlates with human judgment or perceived quality. This weakens the claim “the metric enables ......, supporting consistent ranking”.

In the paper, only 6+1 scenarios are mentioned but in Table 1, it writes 7+1. the instructions of other benchmarks are far more than the proposed one. Indeed, the 7Bench only contains 244 samples, however, other methods have for example, 24k and 45k. Moreover, the number of scenarios is similar to number of samples. It leads confusion for your novelty and contribution.

There’s no ablation experiments.

## From the perspective of writing

In abstrcat, mentioned detailed data of performance, too specific for abstract.

In introduction, the paper mentions the experiment results. For sentence in line 91-93, there’ s no need to directly deliver the experiment results in introduction and comments on tested models are not the purpose of this research.

The whole last paragraph in introduction section is more like result or conclusion and could be excluded from introduction section.

Poor literature review.

In section 2.1, among 6 scenarios, wrong spell “object biding”, binding instead.

In line 268, v(i)∈C seems more like v(i)∈D.

In line 285 “we are also gives”, grammar mistake.

From Figure 2 to Figure 5, although it is easy to distinguish what those abbreviations (e.g. M, G, G_BD) mean, it is better to briefly clarify.

In the paper, 7Banch is frequently mentioned and taken as comparison with the proposed one, while other benchmarks mentioned in Table 1 are only mentioned in the table.

**Questions:**

With 8 samples per instruction, confidence intervals could be reported. Are the differences between models statistically significant?

---

### Official Review · Reviewer_A624 · 2025-10-29

**Soundness:** 2
**Presentation:** 3
**Contribution:** 2
**Rating:** 2
**Confidence:** 4

**Summary:**

The paper proposes a scalable evaluation suite for layout‑guided text‑to‑image generation comprising: 1) C‑Bench (closed‑set) with 3,328 prompt-layout pairs automatically generated via a Prompt Generation Engine (template + LLM) and a Layout Generation Engine (constraint‑based box synthesis). They target specific capabilities such as objects, color, attribute binding, object relations, small boxes, overlapped boxes, and a complex composition scenario (LLM‑generated prompts). 2) O‑Bench (open‑set) where prompts and human boxes curated from Flickr30k Entities test set (down‑sampled to 3,319 prompts for compute), offering natural language and real‑world layouts. The authors generate 8 images per prompt for robustness. They unify the text and layout alignment metrics into a unified score. They evaluate six models (Stable Diffusion v1.4, Cross‑Attention Guidance, GLIGEN, Attention Refocusing, BoxDiff, and MIGC) and produce ~319k images across both benches. MIGC ranks highest (C‑Bench: 0.7082, O‑Bench: 0.7548; Table 4, p. 8).

**Strengths:**

- Easy to follow and clear to understand
- Details about datasets, models, settings and compute are clearly mentioned
- Scalable benchmarks proposed in the paper is a good contribution to the community.

**Weaknesses:**

- Metric/tool dependence remains under-analyzed. Both metrics (text and layout) use third-party models, TIFA for semantics and OWLv2 for spatial grounding and the pipeline hard-filters detections by exact label match before picking the top box. There’s no stress-test with alternative VQA/detectors or synonym/lemmatization handling, so rankings may reflect some biases.
- Unified score may obscure diagnosis. Collapsing two distinct metrics into a single number reduces interpretability when comparing systems. The chosen harmonic mean couples text and layout metrics. If one term is high and the other low, the unified score is driven down (it is proportional to the product over the sum), masking which capability needs improvement.
- Closed-set layout synthesis lacks 3D/occlusion semantics. In Sec. 2.2, single-object boxes are placed uniformly at random. For multi-object prompts, placement varies with relations or non-overlap/overlap rules. There is no explicit depth/ordering policy (which object should be visually front/back), no occlusion reasoning, and no discussion of how draw order (small-first vs. large-first) might change generation outcomes, especially in the overlapping scenario where later boxes are forced to intersect earlier ones. This could bias both images and scores.
- There seems to be ambiguities in box policies. The specification does not say which boxes are instantiated first under overlap (e.g., largest-first to anchor or smallest-first for visibility), nor how near-misses or rejection retries are bounded. With random placement for single objects and relation-driven but heuristic placement for multiple objects, the benchmark can introduce layout artifacts that models learn to game. Also why is it necessary for the single objects to be bigger? If we take depth into the account, a single object could be placed far off in the picture, potentially rendering it small in the ideal generated image.
- I believe some related works and comparisons are missing. Important contemporaries such as LLM-Blueprint [1] and LLM-Grounded Diffusion [2] are not cited or evaluated.
- Calibration of the unified score is unproven. The method assumes text and layout scores are on comparable scales/variances; there’s no evidence they are, nor an exploration of alternate fusions (e.g., weighted HM, min-operator for strict gating). Model rankings may shift under reasonable re-weightings.
- The use of external-model ground truth is sub-optimal. Because both metrics derive correctness from black-box models (LLM-generated Q/A for TIFA; OWLv2 detections), errors or biases in those components directly affect final scores. A small human-rated subset to report human metric correlation is necessary.

Minor:
[Line 259] "text-alignment" --> "layout alignment score”

References

[1] H. Gani et al. "LLM Blueprint: Enabling Text-to-Image Generation with Complex and Detailed Prompts". ICLR 2024

[2] L. Lian et al. "LLM-grounded Diffusion: Enhancing Prompt Understanding of Text-to-Image Diffusion Models with Large Language Models". TMLR 2024.

**Questions:**

- For Label matching with OWLV2, Do you lemmatize or synonym‑map object names before filtering detections?
- How robust are the metrics? In other words, how sensitive are rankings to the choice of VQA model (for TIFA) and detector (for layout)?
- How do you handle prompts with ambiguous relations or multi‑label referents in O‑Bench (e.g., “men” vs “man” or similarly "a dog and cat sitting on each side" with no reference to who is on which side)?

---

### Official Review · Reviewer_cNi8 · 2025-10-31

**Soundness:** 2
**Presentation:** 2
**Contribution:** 2
**Rating:** 2
**Confidence:** 3

**Summary:**

This paper provides a scalable evaluation benchmark of both closed-set and open-set evaluation of layout-to-image generation. The closed set combines the template- and LLM-based prompt generation, which is designed to isolate object attributes. The open-set is curated from FLickr30k Entities to reflect real-world variability. Extensive evaluation has been done across six models in terms of the unified metric they proposed.

**Strengths:**

- Good coverage of the proposed benchmark including both closed- and open-set.

- The proposed unified metric enables consistent model ranking, but still allows breakdown by semantic and spatial perspectives.

- The reproducibility is good. The benchmarks, code, and metrics will be released soon, which is impactful for this community.

**Weaknesses:**

- Missing sub-sessions in Session 3 open-set benchmark. In this session, it only has 3.1 data curation, ends with missing illustration of layout generation, object-box association, etc. I think what has been missed here is very important to understand the pipeline of constructing open-set benchmark.

- The novelty of generating data is limited. For closed-set construction, it is more like a combination of previous template-based and prompt-based generation approach. For open-set, the lay-out generation is not even stated. How to scale the generation of open-set data is one of the major question that readers care about.

- No justification of the unified score. As one of the contribution, it is reasonable to use harmonic mean to penalize imbalances of two components. But it may over-penalize it, and need justification on the sensitivity over other choices such as weighted sum and geometric mean.

**Questions:**

- Can you also add the close/open set type of previous benchmarks?  Readers may be interested in a certain type.

- Open-set curation discard "outliers". It is a good choice for runtime, but may miss the challenging long-tail prompts that should also be considered in a benchmark.

---

### Official Review · Reviewer_j9vm · 2025-10-31

**Soundness:** 3
**Presentation:** 2
**Contribution:** 3
**Rating:** 6
**Confidence:** 4

**Summary:**

This paper presents a comprehensive benchmark and evaluation framework for assessing layout-guided text-to-image diffusion models, which must satisfy both semantic and spatial constraints. The authors propose two complementary benchmarks:
a closed-set benchmark and an open-set benchmark. They further propose a unified metric (sunified) combining semantic alignment (TIFA-based) and spatial alignment (IoU-based AUC) via the harmonic mean, thus allowing consistent ranking across models.

**Strengths:**

1. The work targets a core gap: standardized evaluation of layout-guided text-to-image diffusion models.

2. The combination of closed-set (controlled) and open-set (real-world) evaluation is elegant and complementary.

3. The harmonic-mean–based unified score provides interpretable and fair model ranking.

**Weaknesses:**

1. The unified score assigns equal weight to the semantic and spatial components, but no sensitivity analysis is provided. In practice, certain tasks may place greater emphasis on spatial accuracy than on semantics. Introducing a weighted or learned weighting scheme could enhance the interpretability and flexibility of the metric.

2. While the C-Bench templates are well designed, they are still limited to configurations with up to four objects and predefined spatial relations. More complex forms of compositional reasoning—such as nested spatial relations or counting—remain unexplored.
In addition, it is unclear why a separate closed-set benchmark is necessary if the open-set already encompasses similar contents. The paper should clarify the complementary roles or distinct evaluation purposes of these two settings.

3. The comparison with layout-aware generation benchmarks omits several recent works in the layout-to-image generation literature, such as CreatiLayout and HicoNet. Moreover, the benchmark comparison section does not include the most recent evaluation protocols, e.g., LayoutSAM-Eval, which would provide a fairer and more up-to-date reference.

**Questions:**

1. How sensitive is the unified score to the detector or VQA model used? Could replacing OWLv2 or TIFA significantly alter rankings?

2. Have you compared the unified score with human judgments on a small subset to confirm its validity?

3. Could this framework be generalized to region-conditioned image editing or 3D layout generation tasks?

4. How are ambiguous prompts handled in C-Bench (e.g., “a small cup next to a large cup”)—is there any disambiguation mechanism?

5. For O-Bench, did you perform data leakage checks to ensure models were not trained on Flickr30k Entities?

6. Would the authors consider releasing evaluation scripts as part of an automated leaderboard to encourage consistent reporting?

---

### Note · Authors · 2025-11-13

**Comment:**

We thank the reviewers for their valuable feedback. The authors have decided to withdraw the submission.

**Withdrawal Confirmation:**

I have read and agree with the venue's withdrawal policy on behalf of myself and my co-authors.